# Identification of Novel Mosaic Variants in Focal Epilepsy-Associated Patients’ Brain Lesions

**DOI:** 10.3390/genes16040421

**Published:** 2025-03-31

**Authors:** Camila Araújo Bernardino Garcia, Muhammad Zubair, Marcelo Volpon Santos, Sang Hyun Lee, Ian Alfred Graham, Valentina Stanley, Renee D. George, Joseph G. Gleeson, Hélio Rubens Machado, Xiaoxu Yang

**Affiliations:** 1Department of Surgery and Anatomy, Ribeirão Preto Medical School, University of São Paulo (USP), Ribeirao Preto 14049-900, SP, Brazil; camilaaraujo@usp.br (C.A.B.G.); marcelovolpon@usp.br (M.V.S.); 2Department of Human Genetics, Utah Center for Genetic Discovery, University of Utah, Salt Lake City, UT 84112, USA; u6062200@utah.edu (M.Z.); u6063942@utah.edu (S.H.L.); u0678235@utah.edu (I.A.G.); 3Department of Neuroscience, University of California, San Diego, CA 92093, USA; vstanley@health.ucsd.edu (V.S.); reneegeorge@gmail.com (R.D.G.); jogleeson@health.ucsd.edu (J.G.G.); 4Rady Children’s Institute for Genomic Medicine, San Diego, CA 92123, USA

**Keywords:** focal cortical dysplasia type III, epilepsy, whole-exome sequencing, somatic variants

## Abstract

Focal cortical dysplasia type III (FCDIII) is a rare and complex condition associated with drug-resistant epilepsy and often characterized by cortical lamination abnormalities, along with a variety of neoplasms and vascular abnormalities. Objectives: This study aimed to elucidate the genetic architecture underlying FCDIII through the use of whole-exome sequencing (WES) of brain and peripheral blood samples from 19 patients who had been diagnosed with FCDIII. Methods: Variants were identified through a series of machine-learning-based detection and functional prediction methods and were not previously associated with FCDIII. Mosaic fraction scores of these variants validated the variants’ pathogenicity, and in silico and gene ontology enrichment analyses demonstrated that these variants had severe destabilizing effects on protein structure. Results: We reported ten novel pathogenic somatic missense and loss of function variants across eight genes, including *CNTNAP2*, *ACY1*, *SERAC1*, and *BRAF*. Genetic alterations were linked to clinical manifestations, such as encephalopathies and intellectual disabilities, thereby emphasizing their role as molecular drivers of FCDIII. Conclusions: We demonstrated that next-generation sequencing-based mosaic variant-calling pipelines are useful for the genetic diagnosis of FCDIII, opening up avenues for targeted therapies, yet further research is required to validate these findings and examine their therapeutic implications.

## 1. Introduction

Focal cortical dysplasia (FCD) is one of the primary causes of drug-resistant focal epilepsy, and currently, surgical resection and laser ablation are the only effective treatments. FCD lesions typically are heterogeneous, exhibiting variations in cytoarchitectural abnormalities with regions of higher grade lesion noted, correlated with higher fractionated low-grade lesions [1,2].The distinction between type I, II, and III focal cortical dysplasia (FCD) is pivotal for diagnosis and management. Type I FCD involves isolated architectural abnormalities (Ia/Ib/Ic) without dysmorphic neurons/balloon cells, while type II (IIa: dysmorphic neurons; IIb: +balloon cells) shows cytological disruptions and MRI markers like transmantle signs [3,4]. Type III FCD is defined by association with principal lesions (e.g., hippocampal sclerosis, tumors) and dual pathology, contrasting with the isolated types I/II [5,6]. Clinically, type II exhibits higher epileptogenicity and better surgical outcomes than types I/III, necessitating integrated diagnostic approaches [7,8]. FCD type III is a rare malfunction that demonstrates cortical lamination abnormalities by histopathological analysis [9]. FCD subtypes are diagnosed based on distinct cortical abnormalities that often underlie drug-resistant epilepsy. Among these subtypes, FCDIIIa is associated with hippocampal sclerosis-like tuberous sclerosis, while FCD type IIIb is a common subtype and is specifically associated with developmental or tumor-related lesions affecting the cortex, such as glioneuronal tumors (e.g., ganglioglioma, dysembryoplastic neuroepithelial tumor [DNT], and papillary glioneuronal tumor [PGNT]). FCD type IIIc is associated with vascular malformations, such as Sturge–Weber syndrome. In contrast, the FCD type IIId subtype is characterized by cortical dysplasia co-occurring with early developmental lesions, such as those observed in genetic disorders like diffuse astrocytoma grade II, oligoastrocytoma grade II, ganglioglioma grade I, dysplastic gangliocytoma of the cerebellum (rare tumor), ganglioma, and low-grade neuroepithelial neoplasm [10]. As a subgroup of focal cortical dysplasias, FCDIII presents diagnostic and therapeutic difficulties, which often require surgical treatments because of its amenability to recurrent seizures [11]. This condition is particularly characterized by the absence of normal cortical layers and is accompanied by additional pathological changes, such as neoplasms or abnormal blood vessel formations that must be identified by histopathological methods [12]. The present study highlights the need for advanced comprehension of the elements involved in FCDIII regarding the genetic factors and somatic mutations, such as the pathogenic somatic mutations and the mosaic germline mutations [13].

Deep next-generation sequencing on surgically resected tissues, compared to peripheral blood samples or salivary cells from the same patients, has shown somatic mutations in 10–63% of FCD samples [14]. With the advances of next-generation sequencing (NGS) and machine learning-based computational methodologies, these somatic alterations can now be identified, allowing for a better understanding of the genetic architecture underlying this condition [15]. Next-generation sequencing holds promise for furthering the diagnosis of FCDIII and providing innovative treatment strategies [16]. NGS allows the identification of somatic low-frequency alterations and mosaic mutations that older methods of sequencing may miss. Additionally, an understanding of the interplay between somatic mutations vs. germline mosaicism may inform clinical decision-making regarding FCDIII, a condition in which genetic insight is still central to the challenges presented. Ongoing research elucidating relationships among somatic mutations, mosaic variant germline alterations, and clinical outcomes may provide the opportunity to tailor interventions to patients based on their genetic profiles, which may lead to improved management and care for patients with FCDIII [17]. The neurodevelopment associated with FCDIII is further complicated by mutations in the *PIK3CA* gene, which contribute to abnormal signaling pathways that affect cell growth and differentiation [18].

In this study, we aimed to find candidate genes involved in the FCDIII disease phenotype. We collected samples from FCD lesions, perilesional tissues, and peripheral blood from 19 patients with pathologically confirmed FCDIII. Whole-exome sequencing (WES) was performed, and 11 patients were selected for further analyses to screen for potentially deleterious somatic variants that were exclusively found in FCDIII lesions, followed by in silico bioinformatics analysis for validation [19]. We found 10 potentially pathogenic somatic variants: contactin-associated protein 2 (*CNTNAP2),* aminoacylase 1 (*ACY1*), serine active site containing 1 (*SERAC1*), zinc finger protein 479 (ZNF479), B-Raf proto-oncogene serine/threonine-protein kinase (*BRAF*), (phosphatidylinositol glycan anchor biosynthesis class O) PIGO, tetratricopeptide repeat and ankyrin repeat containing 1 (*TRANK1*), MutS homolog 6 (*MSH6*), ankyrin repeat domain 40 (*ANKRD40)*, and NK2 homeobox 2 *(NKX2-2)*. These genes were not previously associated with FCDIII malformations. We also found somatic mutations in *ACTN2*, *MYO7A*, *LRPPRC*, *ALMS1*, *LRP2*, *PCDHB11*, *GAB4RA1*, and *UNC13C*, which were previously reported to be associated with related phenotypes. *PIK3CA* is prominently linked to FCDIII via mTOR pathway dysregulation, and current studies suggest that previously reported genes with related phenotypes may underscore their recurrent roles in cortical malformations. These findings suggest shared mechanisms (e.g., cytoskeletal dysregulation, synaptic signaling, neuronal migration or cortical layering, and synaptic transmission) [4] across FCDIII subtypes, with *PIK3CA* acting as a central driver. The recurrence of these genes reinforces their potential as modifiers or secondary contributors, warranting deeper functional studies to delineate their precise roles in FCDIII pathogenesis.

## 2. Materials and Methods

### 2.1. Recruitment and Cohort Profile

The 19 subjects of this study were enrolled with specific IDs (Appendix A) at the University Hospital of Ribeirão Preto Medical School (HCFMRP-USP) after detailed clinical investigation and after being diagnosed with focal dysplasia type III (FCDIII) (Appendix A). Following an extensive presurgical evaluation that included magnetic resonance imaging (MRI) and video electroencephalography (EEG) (Appendix A), the surgical strategy was determined to accurately localize the epileptogenic zone. The surgical procedure was proposed based on the clinical and electroencephalographic data when a focal, regional, or hemispheric epileptogenic zone was identified. For sequencing purposes, venous blood samples were collected at the time of surgery, as well as the biopsied tissue, in order to detect the mosaic variants.

### 2.2. Whole-Exome Sequencing and Data Analysis

The exome sequencing was performed to detect the disease-causing mosaic variants, and detailed processes are shown in Figure 1. Quality control and pre-processing of the sequencing data were performed using FastQC [20], MultiQC [21], and TrimGalore (https://www.bioinformatics.babraham.ac.uk/projects/trim_galore/, accessed on 19 September 2024). After pre-processing, the reads were aligned to the human reference genome (GRCh38 or GRCh37) using the Burrows–Wheeler Aligner [22] with the BWA-MEM algorithm. Post-alignment, coverage metrics, and duplicate analysis were conducted using Picard’s MarkDuplicates (https://broadinstitute.github.io/picard/, accessed on 20 September 2024). Variant calling was executed using Mutect2 [23], generating variant call format (VCF) files. Subsequently, these variants were annotated and functionally characterized using SnpEff [24], the Universal Mutation Database [25], and custom in-house Perl scripts. These variants were therefore filtered (initially) down to those with minimum coverage of ≥30 in normal tissue for the annotated variants. To prioritize pathogenic variants, we identified potential effects of the variants, including missense variants, splice donor/acceptor variants, stop gained/lost variants, start lost variants, TF binding site variants, 5′ UTR premature start codon gain variants, and stop retained variants, using the reference genome mentioned above. Patients with positive variants are supparmized in Table 1.

### 2.3. Somatic Variants Identification and Mosaic Fraction Calculation Methodology

The variants were functionally annotated and filtered using VEP version 111 and ANNOVAR version 2019Oct24 to identify disease-causing variants. The tools integrated the various built-in public databases, such as Online Mendelian Inheritance in Man (OMIM), InterVar, ClinVar, the Human Gene Mutation Database (HGMD), dbSNP, 1000 Genomes Project (1000G), Exome Sequencing Project (ESP), ExAC, gnomAD, and HGMD Professional databases. Variants meeting pathogenicity criteria, as supported by CADD scores, population allele frequency, POLYPHEN predictions, and SIFT scores, were retained, and deep-learning-based methods, such as GPN-MSA (Table 2, Appendix A), were also used to predict the consequences of the variants; variants failing to meet these criteria were removed. The detailed filtration process is described in Figure 1, Appendix A. The mosaic fraction was calculated using the variant allele frequency (VAF), determined as follows: VAF = (ALT Reads in Tumor)/(Total Reads in Tumor) and then Mosaic Fraction = 2 × VAF. Somatic mutation confirmation was based on the absence of ALT reads in normal samples (ALT = 0), indicating that the mutation was acquired and not germline (Table 2, Appendix A).

### 2.4. Bioinformatics In Silico Analysis

To evaluate the functional and structural impact of identified variants, we conducted bioinformatics in silico analyses. First, gene ontology and pathway enrichment were assessed using Metaspace [26] to prioritize biologically relevant genes. Next, variants were analyzed for protein stability and pathogenicity using AlphaFold (predicted 3D structures) [27] and DDMut (free energy change [ΔΔG, kcal/mol], with lower ΔΔG indicating destabilization) [28]. Variants with ΔΔG ≤ −kcal/mol were classified as destabilizing, aligning with established thresholds for pathogenic impact.

### 2.5. Data Analysis and Visualization

All the data were evaluated and visualized by using R Studio version 4.3.3 and R packages like ggplot2, ggseq, and dplyr.

## 3. Results

### 3.1. Clinical Characteristics of FCDIII Patients in This Study

For this study, 19 subjects, all diagnosed with focal cortical dysplasia type III (FCDIII), were recruited to identify disease-causing mosaic variants (Figure 1). Epilepsy onset occurred at a median age of 1 year and 3.5 months, with monthly seizure frequencies of 4 to 1200. Seizure types were documented using comprehensive clinical assessments consisting of symptomatic focal, focal, generalized tonic–clonic, generalized, and structural focal. Table 1 and Appendix A provide additional clinical parameters, including ADNPM, neuropsychiatric impairment, hemiparesis, ocular deviation nystagmus, and macrocephaly, for each patient. The mean age at the time of surgery was 7 years and 10 months. All subjects underwent complete clinical histories, and both preoperative and postoperative brain imaging (EEG and MRI) supported the FCDIII phenotype in the cohort. Based on preoperative pathological evidence of lesions on imaging, the biopsy was performed on brain tissue samples and confirmed intraoperatively using a neutralization stealth system. Cortical resection procedures were performed for refractory epilepsy in eight participants aged 1 to 13 years at the time of surgery (Table 1 and Appendix A), providing detailed observations about tissue sampling and lesion confirmation. Based on the lesions’ pathology, patients were diagnosed and categorized into subtypes: FCDIIIa had 3/19 patients with tuberous sclerosis and dysplastic cortical focal epileptogenic abnormality; FCDIIIb had 2/19 patients with DNET, focal cortical dysplasia, ganglioglioma, EMT, and DNET; FCDIIIc had 1/19 patient with Sturge–Weber syndrome and dysplastic cortical focal epileptogenic abnormality; and FCDIIIDd had 13/19 patients with diffuse astrocytoma grade II, oligoastrocytoma grade II, ganglioglioma grade I, dysplastic gangliocytoma of the cerebellum (a rare tumor), ganglioma, and low-grade neuroepithelial neoplasm (Appendix A and Figure 1A).

### 3.2. WES Analyses Identified Ten Novel Potential Pathogenic Candidate Variants in Eight FCDIII Patients

We performed whole-exome sequencing (WES) on both the blood and brain biopsy samples used from FCDIII patients to identify pathogenic mosaic variants. Pairing DNA samples from the brain and blood were sequenced to understand the role somatic variants play in mediating the underlying FCDIII phenotype (Figure 1). Whole-exome sequencing (WES) was performed on paired brain–blood samples from 19 FCDIII patients. After quality control and variant filtration (Figure 1A), all patients harboring high-confidence somatic variants were selected for detailed analysis (Table 1, Figure 2B). Of these, eight patients exhibited pathogenic variants meeting functional and clinical thresholds (Figure 1B). The WES data were then processed with extensive variant filtration pipelines, as shown in Figure 1. The variants were prioritized with frameshift mutations, inframe deletions, missense mutations, stop-gain mutations, splice acceptor variants, or splice donor variants if the protein variant had been previously identified to be pathogenic in functional studies or if the variant was associated with FCDIII or related syndromes. Subsequently, we filtered using pathogenic criteria like CADD, SWIFT, POLYPHEN, and genomic frequency, and we found 35 pathogenic somatic variants.

We considered variants that affected the protein structure/transactions based on this filtration criteria and found 10 potential disease-causing variants in eight patients (Table 2). A total of 6 were found to be missense pathogenic variants (Figure 2B,C); they were (HR83) *CNTNAP2*, ENST00000361727.3:c.1220A>G and ENSP00000354778.3:p.Asn407Ser; (HR83) *ACY1*, ENST00000404366.2:c.699A>C and ENSP00000384296.2:p.Glu233Asp; and (HR83) *SERAC1*, ENST00000367104.3:c.89T>C and ENSP00000356071.3:p.Ile30Thr, associated with FCDbIII in the HR83 patient. *ZNF479*, ENST00000319636.10:c.974G>A and ENSP00000324518.6:p.Cys325Tyr were identified in HR100 patients with FCDIIIb. *BRAF*, ENST00000646891.2:c.1799T>A and ENSP00000493543.1:p.Val600Glu were identified in HR139 and HR192 patients with FCDdIII; *PIGO*, ENST00000378617.4:c.2432G>A and ENSP00000367880.3:p.Arg811Gln were also associated with FCDIIId in HR192 patients. The four potential loss of function/pathogenic frameshift shift truncating variants were found in four patients. *TRANK1*, *ENST00000645898.2:c.2798_2799insACCACCGA* and *ENSP00000494480.1:p.Ile934ProfsTer16** frameshift variants were identified *in* HR94 associated with FCDIIIa; *MSH6* splice_acceptor_variant, ENST00000234420.11:c.3557-3_3557del in HR(100) with FCDIIIa was found in HR185 associated with FCDIIIb; and for the splice_acceptor_variant *ANKRD40*, *ENST00000285243.7:c.779-2_784del* and *NKX2-2*, *ENST00000377142.4:c.370del* and *ENSP00000366347.4:p.Asp124ThrfsTer60**, a frameshift was found in HR175 and 175 associated with FCDIIId (Figure 1 and Table 2).

The mosaic fraction score of each pathogenic variant was assessed to validate their pathogenicity. The *CNTNAP2* c.1220A>G variant showed a mosaic fraction of 62%, suggesting subclonal expansion. The *ACY1* c.699A>C variant had a high mosaic fraction of 97.72%, indicating that it is likely a dominant mutation within the tumor and may have arisen early in tumor development. The *SERAC1* c.89T>C variant displayed a mosaic fraction exceeding 100% (107%), suggesting tumor-specific amplification, which could be due to either a gain of the mutated allele or a loss of the wild-type allele. Additionally, the mosaic fraction scores for *ZNF479* c.974G>A (11%) and *BRAF* c.1799T>A (18% in HR139 and 54% in HR192) indicated subclonal expansion and moderate clonal expansion. Similarly, the variants *PIGO* c.2432G>A (5%), *TRANK1* c.2798_2799insACCACCGA (5%), *MSH6* c.3557-3_3557del (14%), *ANKRD40* c.779-2_784del (19%), and *NKX2-2* c.370del (4%) also suggested clonal expansion (Table 2).

Furthermore, we also identified several pathogenic variants with clonal status in tumors, based on their mosaic fraction scores. The variants *LRPPRC*, *MYO7A*, *ACY1*, and *ABHD14A-ACY1* exhibited mosaic fractions of 102.4%, 98.8%, 97.5%, and 97.5%, respectively, suggesting they may be disease-causing somatic mutations. Additionally, *ALMS1* and *LRP2* displayed mosaic fractions of 75% and 85%, respectively (Appendix A), indicating clonal progression. Additionally, GPN-MSA_SCORES for the following variants indicated high pathogenic potential: *PCDHB11I* (−9.39), *GAB4RA1* (−8.55), and *UNC13C* (−7.9), all of which have been previously associated with neurological disorder-related phenotypes (Appendix A). Overall, these findings support the pathogenicity of these mosaic variants.

### 3.3. The Novel Identified Pathogenic Mosaic Variants That Showed Phenotypic Consistency and Severe Protein Structural Damage

The gene ontology enrichment analysis showed how the identified missense variants mosaics were strongly associated with severe encephalopathies and intellectual disabilities, as indicated by high −log_10_ (*p*) [26] values (Figure 3A). These findings therefore directly linked genetic variants to patient clinical manifestations, and enriched terms, such as encephalopathies, brachycephaly, and nasal bridge anomalies, suggested an active role in disease etiology (Figure 3B). This analysis highlighted the fundamental contribution of these variants to the patient’s phenotypic abnormalities. In silico analysis was then carried out to evaluate the impacts of these variants on protein structure and stability to further investigate pathogenicity.

Consistent with their severe functional consequences, destabilizing effects were significant for all variants (Figure 4). Key findings included (HR83) CNTNAP2, ENSP00000354778.3:p.Asn407Ser, which exhibited a stability impact of −0.56 kcal/mol; (HR83) ACY1, ENSP00000384296.2:p.Glu233Asp, with −1.25 kcal/mol; *SERAC1*, ENSP00000356071.3:p.Ile30Thr, with −0.22 kcal/mol; and (HR192&HR139) BRAF, ENSP00000493543.1:p.Val600Glu, showing −0.84 kcal/mol. Similarly, the identification of loss of function/frameshift variants (HR94) TRANK ENSP00000494480.1:p.Ile934ProfsTer16, (HR100) MSH6, (HR185) ANKRD40, (HR175) NKX2-2, and ENSP00000366347.4:p.Asp124ThrfsTer60 may contribute to the disease mechanism by insertion of a premature stop codon that leads to truncation and disrupts the functional integrity of the protein.

These destabilizations indicate that the variants interfere with protein folding or function, further implicating them in the disease process. The results of these structural and gene ontology analyses coupled provide compelling evidence that these mosaic missense variants are damaging. The accumulation of phenotypic and molecular evidence is consistent with the interpretation that these variants are important drivers of the patient’s disease. These results demonstrate how these mosaic variants are functional and structural drivers of the patient’s severe clinical phenotype. As a result, this suggests that their pathogenic potential is more marked for their association with severe intellectual disabilities and with protein instability and that precisely investigating their biological mechanisms and even their possible therapeutic targets are to be further performed.

The functional significance of each gene variant is represented by −log_10_ (*p*), ranging from −5 to −1, where higher values indicate greater functional relevance in the disease condition.

## 4. Discussion

Focal cortical dysplasia type III (FCDIII) represents a rare and challenging subtype of cortical dysplasias associated with drug-resistant epilepsy, necessitating surgical intervention [29]. This condition is typified by cortical lamination abnormalities, often accompanied by additional pathological changes, such as neoplasms or vascular malformations [30,31]. Unlike FCDII (mTOR-associated, e.g., *PIK3CA/AKT3)* mutations [32] affected by the mTOR pathway, FCDIII exhibits genetic heterogeneity (*LRPPRC*, *UNC13C*) and dual pathology (tumors/vascular malformations) with indirect mTOR linkage because mTOR drives FCDII cytopathology (e.g., balloon cells), and FCDIII pathogenesis involves broader somatic and mosaic mechanisms. This distinction underscores the necessity of advanced genetic profiling to disentangle FCDIII-specific pathways [31]. FCD IIIa is associated with hippocampal sclerosis, FCDIIIb with tumor-related lesions (e.g., ganglioglioma, DNT, and PGNT), FCDIIIc with vascular malformations (e.g., Sturge–Weber syndrome), and FCDIIId with early developmental lesions seen in genetic disorders (e.g., diffuse astrocytoma, ganglioglioma, and low-grade neuroepithelial neoplasms). The study’s results emphasize the critical role of advanced genetic approaches, such as next-generation sequencing (NGS) [33], in uncovering somatic and mosaic mutations contributing to the pathophysiology [34] of FCDIII.

Overall, participants with disease-causing somatic variants in neurodevelopmental genes exhibited a range of clinical and neuropsychological phenotypes. One of the significant findings of this research was the identification of 10 pathogenic somatic variants in eight FCDIII patients through whole-exome sequencing (WES).

These findings confirmed the pathogenicity of the identified mosaic variants. The *ACY1* variant (97.72%) likely represents an early dominant somatic mutation, while *CNTNAP2* (62%) and *BRAF* (18%) suggest subclonal expansion. The *SERAC1* variant (107%) indicates tumor-specific amplification. Lower mosaic fractions in *ZNF479*, *PIGO*, *TRANK1*, *MSH6*, *ANKRD40*, and *NKX2-2* further support clonal expansion. Together, these results underscore the roles of these variants in tumor development. These variants were located in genes previously unassociated with FCDIII, such as *CNTNAP2*, *ACY1*, *SERAC1*, and *BRAF*. The discovery of these genetic alterations underscores the complex genetic architecture of FCDIII, wherein both germline and somatic mutations interact to influence disease manifestation. Notably, the mosaic nature of these variants emphasizes the importance of sequencing-affected tissues directly, as opposed to relying solely on peripheral blood samples, which may not capture these low-frequency alterations [35].

The gene ontology analysis [26] further strengthens the connection between the identified variants and the clinical phenotypes observed in FCDIII patients. Variants in genes such as *BRAF* and *CNTNAP2* were associated with severe protein destabilization and phenotypic abnormalities, including encephalopathies and intellectual disabilities [36]. These findings suggest that the pathogenic potential of these variants stems not only from their presence but also from their significant impact on protein function and stability [37]. The identification of pathogenic variants, such as those in *TRANK1*, *MSH6*, *ANKRD40*, and *NKX2-2***,** their association with severe clinical phenotypes, and their molecular consequences represent significant strides toward improving the diagnosis and management of this condition [38].

The correlation between genetic alterations and clinical manifestations highlights the potential for personalized medicine approaches tailored to the specific genetic profile of FCDIII patients. The identification of clonal pathogenic variants through mosaic fraction analysis offers crucial insights into tumorigenesis. Variants in *LRPPRC*, *MYO7A*, *ACY1*, and *ABHD14A-ACY1* showed high mosaic fractions (102.4%, 98.8%, 97.5%, and 97.5%), indicating their potential role as early somatic drivers. These variants are linked to processes like mitochondrial regulation and amino acid metabolism [39], critical for tumor survival. In contrast, *ALMS1* and *LRP2*, with mosaic fractions of 75% and 85%, likely contribute to disease progression [40,41], affecting ciliary signaling pathways. GPN-MSA scores highlighted *PCDHB11*, *GABRA4* [42], and *UNC13C* [43] as high-pathogenicity candidates, suggesting their potential pleiotropic effects. *CNTNAP2* has previously been linked to autism and intellectual disability, *BRAF* to epilepsy, *TRANK1* to schizophrenia, and *LRPPRC/MYO7A* to mitochondrial and ciliary disorders. Our findings highlight potential common pathways, including synaptic dysfunction and mitochondrial deficits, emphasizing overlapping mechanisms between FCDIII and intellectual disabilities. This further indicates a broader phenotypic spectrum of FCDIII. Future studies integrating single-cell sequencing could clarify the mechanistic contributions of these variants. In addition to elucidating the genetic basis of FCDIII, this study highlights the diagnostic and therapeutic potential of NGS. By identifying pathogenic variants that traditional methods might overlook, NGS facilitates a deeper understanding of FCDIII’s molecular mechanisms. This knowledge could pave the way for novel therapeutic strategies, including targeted gene therapies and interventions aimed at stabilizing affected proteins. However, the study also underscores the need for further research to explore the biological mechanisms underlying these genetic alterations and their potential as therapeutic targets.

## 5. Conclusions

In conclusion, the findings of this study provide critical insights into the genetic etiology of FCDIII, highlighting the importance of integrating advanced genetic technologies into clinical practice. The identification of novel pathogenic variants, their association with severe clinical phenotypes, and the elucidation of their molecular consequences represent significant strides toward improving the diagnosis and management of this challenging condition. Future research should focus on validating these findings in larger cohorts and exploring targeted therapies to address the underlying genetic abnormalities in FCDIII patients.

## Figures and Tables

**Figure 1 genes-16-00421-f001:**
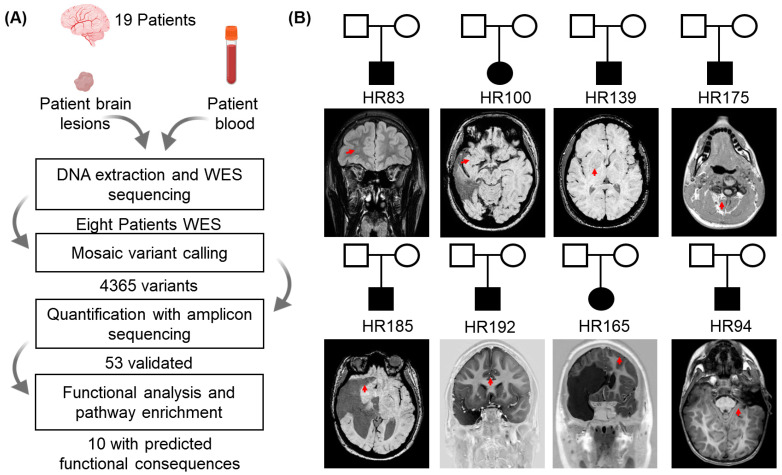
Somatic variants filtering pipeline through WES in FCDII patients. (**A**) Stepwise process and filtering criteria for somatic variants; (**B**) Eight patients from eight distinct pedigrees were diagnosed with focal cortical dysplasia type III (FCDIII). MRI neuroimaging of these patients revealed brain lesions, indicated by red arrows in each image.

**Figure 2 genes-16-00421-f002:**
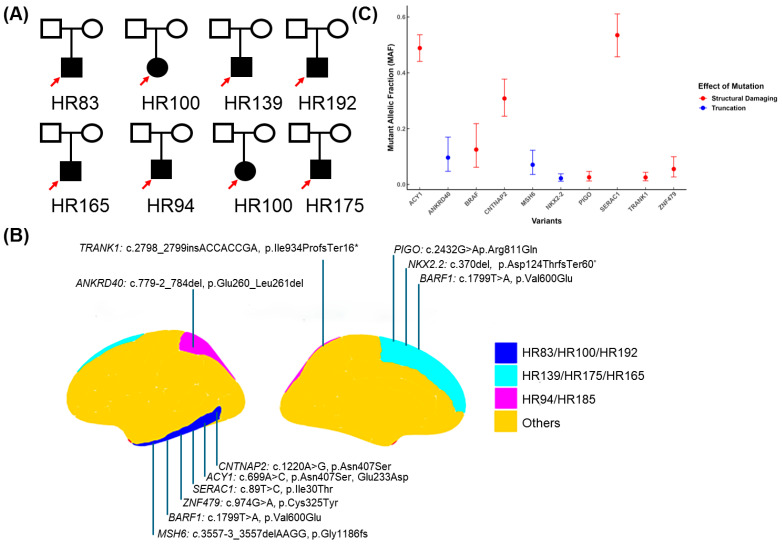
Novel disease-causing somatic variants identified in focal cortical dysplasia patients. (**A**) The red arrow highlights the WES-selected patients in each family. Identified novel somatic variants are displayed below each pedigree, with missense variants shown in red and frameshift variants in blue. (**B**) The identified variants are mapped to specific brain regions, including the temporal, frontal, and multilobar areas. (**C**) Visualization of the damaging effects of these variants, along with their mutant allele fraction (MAF) values; the error bar shows a 95% binomial confidence interval.

**Figure 3 genes-16-00421-f003:**
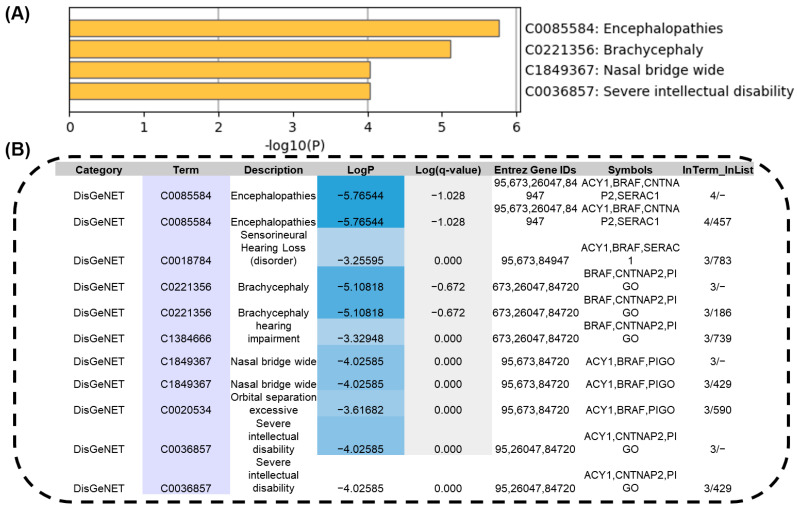
Identified somatic missense variants and their association with disease-related gene ontology through gene enrichment analysis. (**A**) Gene ontology enrichment analysis using MetaScape highlights the functional roles of each novel gene variant associated with the disease condition. (**B**) In Metascape, enrichment analyses scores and significance are reported as −log_10_ (*p*-value). Larger values indicate stronger enrichment to account for multiple comparisons.

**Figure 4 genes-16-00421-f004:**
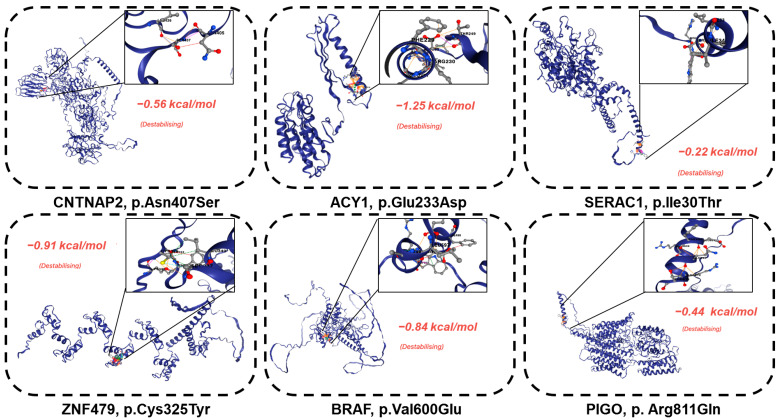
Somatic pathogenic variants destabilize the protein structure. The pathogenic impacts of somatic variants on the protein structures of CNTNAP2 p.Asn407Ser: −0.56 kcal/mol, ACY1 p.Glu233Asp: −1.25 kcal/mol, SERAC1, p.Ile30Thr: −0.22 kcal/mol, ZNF479, Cys325Tyr: −0.91 kcal/mol, BRAFp.Val600Glu: −0.84 kcal/mol, and PIGO, p.Arg811Gln: −0.44 kcal/mol were assessed. The damaging effect was quantified in terms of binding energy change, measured in kcal/mol.

**Table 1 genes-16-00421-t001:** Patients’ clinical characteristics.

ID	HR83	HR100	HR139	HR175	HR185	HR192	HR165	HR94
HOSPITAL ID	1207695H	0769405C	1184599D	1073501B	0905454I	1524999G	1432293H	1286326H
LOBE	RIGHT TEMPORAL	RIGHT/TEMPORAL	RIGHT/FRONTAL	LEFT/FRONTAL	RIGHT/TEMPORAL/PARIETAL	RIGHT/TEMPORAL	LEFT/FRONTAL	RIGHT/MULTILOBAR
WES	BLOOD/FROZEN TISSUE	BLOOD/FROZEN TISSUE	BLOOD/FROZEN TISSUE	BLOOD/FROZEN TISSUE	BLOOD/FROZEN TISSUE	BLOOD/FROZEN TISSUE	BLOOD/FROZEN TISSUE	BLOOD/FROZEN TISSUE
Sex	Male	Female	Male	Male	Male	Male	Female	Male
Date of Birth	12/19/1997	2005/8/3	03/29/2007	03/26/2013	09/14/2005	12/21/2006	09/13/2010	01/14/2012
Date of Surgery	2010/4/7	03/24/2014	11/16/2015	09/28/2015	2017/2/8	09/18/2017	11/30/2017	2013/4/10
Onset of Epilepsy	5 years	7 months	4 years	4 months	7 months	8 years	2 years	2 months
Monthly Seizure Frequency/Pre-operative	90	900	2.5	450	60	540	2000	90
Type of Seizure	Focal	Generalized tonic–clonic	Structural focal	Generalized	Generalized	Symptomatic focal	Structural focal	Focal
ADNPM	N	N	N	Y	Y	N	N	Y
Neuropsychomotor Impairment	N	Y	Y	Y	Y	Y	Y	Y
Hemiparesis	N	N	N	N	N	N	Y	N
Ocular Deviation Nystagmus	N	N	N	N	N	N	Y	Y
Macrocephaly	N	N	N	N	N	N	N	N
Number of Medications	2	3	2	3	3	1	4	2
Treatment Reoperation	N	Y	N	N	N	N	N	N
Engel 1 Year	1	4	1	3	1	3	1	NO DATA
Engel 5 Years	1	4	1	0	NO DATA	NO DATA	NO DATA	NO DATA

**Table 2 genes-16-00421-t002:** Somatic missense variant was predicted to be highly pathogenic due to its damaging.

Mosaic Missense Pathogenic Variants															
ID	Gene	FCDIII Type	Location	Mutation	Mosaic Fractions	HGVSc	HGVSp	SIFT	PolyPhen	CADD PHRAD	GPN-MSA_SCORES	gnomADe_AF	Transcript_ID	Reference_Genome	Gene_ENST
HR83	CNTNAP2	FCDIIIb	7:147132381-147132381	missense variant	62%	ENST00000361727.3:c.1220A>G	ENSP00000354778.3:p.Asn407Ser	tolerated (0.73)	benign (0.01)	8.396	−12.63	0.0006432	ENST00000361727	GRCh38	CNTNAP2-201
HR83	ACY1	FCDIIIb	3:51987188-51987188	missense variant	97.72%	ENST00000404366.2:c.699A>C	ENSP00000384296.2:p.Glu233Asp	deleterious (0.01)	probably_damaging (0.95)	22.6	−5.98	0.00004446	ENST00000404366	GRCh38	ACY1-201
HR83	SERAC1	FCDIIIb	6:158158275-158158275	missense variant	100%(CNV)	ENST00000647468.2:c.89T>C	ENSP00000496731.1:p.Ile30Thr	deleterious (0.01)	benign (0.05)	17.95	−1.63	0.0009691	ENST00000647468	GRCh38	SERAC1-217
HR100	ZNF479	FCDIIIb	7:57120441-57120441	missense variant	11%	ENST00000319636.10:c.974G>A	ENSP00000324518.6:p.Cys325Tyr	tolerated_low_confidence (1)	benign (0)	0.02	−4.3	0.0001944	ENST00000319636	GRCh38	ZNF479-201
HR139	BRAF	FCDIIId	7:140753336-140753336	missense variant	18%	ENST00000646891.2:c.1799T>A	ENSP00000493543.1:p.Val600Glu	deleterious (0)	probably_damaging (0.963)	29.8	−12.63	3.161 × 10^−12^	ENST00000646891	GRCh38	BRAF-220
HR192	BRAF	FCDIIId	7:140753336-140753336	missense variant	54%	ENST00000646891.2:c.1799T>A	ENSP00000493543.1:p.Val600Glu	deleterious_low_confidence (0)	probably_damaging (0.935)	29.8	−12.63	6.905 × 10^−7^	ENST00000646891	GRCh38	BRAF-220
HR165	PIGO	FCDIIId	9:35091455-35091455	Missense variant	5%	ENST00000378617.4:c.2432G>A	ENSP00000367880.3:p.Arg811Gln	tolerated (0.2)	benign (0.007)	18.7	−7.39	0.000591	ENST00000378617	GRCh38	PIGO-203
**Mosaic Loss of Function Pathogenic Variants**															
**ID**	**Gene**	**FCDIII Type**	**Location**	**Mutation**	**Mosaic Fractions**	**HGVSc**	**HGVSp**	**SIFT**	**PolyPhen**	**CADD PHRAD**	**GPN-MSA_SCORES**	**gnomADe_AF**	**Transcript_ID**	**Reference_Genome**	**Gene_ENST**
HR94	TRANK1	FCDIIIa	3:36856923-36856924	frameshift_variant	5%	ENST00000645898.2:c.2798_2799insACCACCGA	ENSP00000494480.1:p.Ile934ProfsTer16	-	-	19.45	3.72	0.00000275	ENST00000645898	GRCh38	TRANK1-205
HR100	MSH6	FCDIIIb	2:47805615-47805618	splice_acceptor_variant,coding_sequence_variant,intron_variant	14%	ENST00000234420.11:c.3557-3_3557del	-	-	-	39	3.18	0.00000138	ENST00000234420	GRCh38	MSH6-201
HR185	ANKRD40	FCDIIId	17:50697116-50697123	splice_acceptor_variant,coding_sequence_variant	19%	ENST00000285243.7:c.779-2_784del	-	-	-	35	-	0.00000206	ENST00000285243	GRCh38	ANKRD40-201
HR175	NKX2-2	FCDIIId	20:21512375-21512381	frameshift_variant	4%	ENST00000377142.4:c.370del	ENSP00000366347.4:p.Asp124ThrfsTer60	-	-	35	-	0.0000062	ENST00000377142	GRCh38	NKX2-2-201

## Data Availability

Data are provided as main and Appendix A.

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
