# Peer review of "Identification of Novel Mosaic Variants in Focal Epilepsy-Associated Patients’ Brain Lesions"

_genes, 2025, doi:10.3390/genes16040421_

Round 1

Reviewer 1 Report

Comments and Suggestions for Authors

The manuscript "Identification of Novel Mosaic Variants in Focal Epilepsy Associated Patients' Brain Lesions" by Garcia et al. is well-written. The authors conducted this study with a small cohort of patients with selectively type III FCD. This study and its results are novel and would interest the journal's readership. My points below are all relatively minor, but I believe they would strengthen the article.

Comment 1: The authors have described Type III FCD well in the first paragraph (lines 36-60). However, they need to provide more information in the introduction about Type I and II FCD and how they are distinct from Type III FCD.

Comment 2: The second paragraph described the connection between NGS and FCDIII. However, it shows only the PICK3CA variant only associated with FCDIII. However, the results have shown that only 10 genes were newly identified, and the remaining were previously reported with FCDIII. Therefore, a few more sentences about previously reported genes.

Comment 3: Section 2.4 is unclear. For example, "Therefore, we performed" and "the variants were also assessed...."

Comment 4: Lines 177-178 say 11 patients were selected, which is unclear in Figures 1 and 2. The authors can cite the table of 11 patients were selected for WES analysis or color code those patients.

Other minor comments:

The discussion part font size is different than the rest of the manuscript.

The mTOR-associated FCD type is unclear in lines 294-95.

FCD was already introduced, and adding Focal Cortical Dysplasia in line 296 is unnecessary.

In lines 324-25, A few genes were bolded.

References were missing for lines 319-27.

Linking potentially other IDD phenotypes to the FCD type III phenotypes would help readers understand how these genes are associated with other diseases.

Author Response

Reviewer 1:

The manuscript "Identification of Novel Mosaic Variants in Focal Epilepsy Associated Patients' Brain Lesions" by Garcia et al. is well-written. The authors conducted this study with a small cohort of patients with selectively type III FCD. This study and its results are novel and would interest the journal's readership. My points below are all relatively minor, but I believe they would strengthen the article.

Comment 1: The authors have described Type III FCD well in the first paragraph (lines 36-60). However, they need to provide more information in the introduction about Type I and II FCD and how they are distinct from Type III FCD.

Response: Thank you for the valuable suggestion. We have addressed this feedback by expanding the introduction to provide a clearer distinction between Type I, II, and III focal cortical dysplasia (FCD). The revised section outlines the isolated architectural abnormalities in Type I, the cytological disruptions and MRI markers in Type II, and the dual pathology associated with Type III. We also emphasized the clinical differences, including the higher epileptogenicity and better surgical outcomes of Type II compared to Types I and III, underscoring the need for integrated diagnostic approaches. We greatly appreciate your feedback in helping us enhance the comprehensiveness of the introduction.

Comment 2: The second paragraph described the connection between NGS and FCDIII. However, it shows only the PICK3CA variant only associated with FCDIII. However, the results have shown that only 10 genes were newly identified, and the remaining were previously reported with FCDIII. Therefore, a few more sentences about previously reported genes.

Response: Thank you for your insightful comment regarding the connection between NGS findings and FCDIII. We have addressed this by expanding the discussion to include previously reported genes with related phenotypes, emphasizing their recurrent roles in cortical malformations. The revised section now highlights shared mechanisms such as cytoskeletal dysregulation, synaptic signaling, neuronal migration, cortical layering, and synaptic transmission across FCDIII subtypes, with PIK3CA acting as a central driver. This recurrence underscores the potential of these genes as modifiers or secondary contributors, warranting further functional studies to elucidate their precise roles in FCDIII pathogenesis. We greatly appreciate your suggestion, which helped us provide a more comprehensive context.

Comment 3: Section 2.4 is unclear. For example, "Therefore, we performed" and "the variants were also assessed...."

Response: Thank you for your thoughtful feedback on Section 2.4. We have revised the section to improve clarity and flow. The updated text now clearly outlines the steps taken to evaluate the functional and structural impact of the identified variants. It details the use of gene ontology and pathway enrichment through Metaspace for prioritizing biologically relevant genes, followed by protein stability and pathogenicity analysis using AlphaFold and DDMut. We also specified the ΔΔG threshold for classifying destabilizing variants. We greatly appreciate your suggestion, which helped enhance the readability of this section.

Comment 4: Lines 177-178 say 11 patients were selected, which is unclear in Figures 1 and 2. The authors can cite the table of 11 patients were selected for WES analysis or color code those patients.

Response: Thank you for the valuable suggestion. We have clarified this point by specifying that whole exome sequencing (WES) was performed on 19 FCDIII patients, with high-confidence somatic variant carriers selected for detailed analysis (Table 1, Figure 2B). Additionally, we highlighted that 8 patients met the pathogenic variant criteria (Figure 1B) to enhance clarity and alignment between the text and figures. We greatly appreciate your feedback in helping us improve the manuscript.

Other minor comments:

The discussion part font size is different than the rest of the manuscript.

Response: Thank you for noticing that inconsistency. We have now adjusted the font size in the discussion section to match the rest of the manuscript for uniformity. We appreciate your attention to detail.

The mTOR-associated FCD type is unclear in lines 294-95.

Response: Thank you for your insightful feedback regarding the mTOR-associated FCD type in lines 294-95. We truly appreciate your attention to detail. In response, we have clarified this section to better distinguish FCDIII from mTOR-associated FCDII. We now highlight the genetic heterogeneity of FCDIII (involving LRPPRC and UNC13C) and its dual pathology (tumors/vascular malformations) with indirect mTOR linkage. This revision emphasizes the broader somatic and mosaic mechanisms underlying FCDIII and the importance of advanced genetic profiling to unravel its specific pathways. Thank you once again for helping us improve the clarity of our manuscript.

FCD was already introduced, and adding Focal Cortical Dysplasia in line 296 is unnecessary.

Response: Thank you for the suggestion. We have removed “Focal Cortical Dysplasia” in line 296 to avoid redundancy.

In lines 324-25, A few genes were bolded.

Response: Thank you for the feedback. We have now ensured consistent formatting throughout the manuscript, including the bolded genes in lines 324-25.

References were missing for lines 319-27.

Response: We sincerely appreciate the reviewer’s feedback. References have now been added to support the claims in lines 319–327.

Linking potentially other IDD phenotypes to the FCD type III phenotypes would help readers understand how these genes are associated with other diseases.

Response: We appreciate the reviewer’s suggestion to contextualize the identified genes within broader neurodevelopmental and intellectual disability (IDD) phenotypes. Below is the revised text integrating these connections, with key additions from lines 377-381 : CNTNAP2 has previously been linked to autism and intellectual disability, BRAF to epilepsy, TRANK1 to schizophrenia, and LRPPRC/MYO7A to mitochondrial and ciliary disorders. Our findings highlight potential common pathways, including synaptic dysfunction and mitochondrial deficits, emphasizing overlapping mechanisms between FCDIII and intellectual disabilities. This further indicates a broader phenotypic spectrum of FCDIII.

Reviewer 2 Report

Comments and Suggestions for Authors

The article genes-3549688 - "Identification of Novel Mosaic Variants in Focal Epilepsy Associated Patients’ Brain Lesions" by Bernardino García & coll.

reports about the genetic modifications underlining the human Focal Cortical Dysplasia Type III (FCDIII), as assessed by whole exome sequencing (WES). They report 10 novel pathogenic changes related to the expression of 8 genes. Some missense variants were considered to be highly pathogenic based on the estimated binding energy change.

The article is surely of high interest for future researches in all the fields of brain science. However, some attention should be given to the writing and to the data presentation:

1) In Materials and Methods:

         1a) the authors states that 19 patients were studied, but in fig. 1 only 8 of them appear to have been studied by WES. How the remaining 11 patients contributed to the study?

         1b) there is NO definition of neither the P value that is the base of LogP, nor of log(q-value) both in Fig. 3B.  

         1c) on line 140, how the "The detrimental effect on protein structure was measured in -kg/kcal according to the specified criteria" has been calculated and how it has been related to  "energy binding value" as reported in Fig.4 ?

2) In Results

         2a) on line 8(156): the sentence "The mean age at the time of surgery was 4 years and 5 months." is confusing since on line 2(150) it is specified that "Epilepsy onset occurred at a median age of 1 year and 11 months..." The authors should specify that the sentence on line 8(156) is referring at the 8 patients that appear on Table 1.

         2b) on line 14(161), it is specified that "Cortical resection procedures were performed for the treatment of refractory epilepsy in 161 14 to 28-year-old participants at the time of referral (Table 1 &S4)" However, according to Table 1, the 28 year old patient received surgery in 2010, when he was 13, and the two youngest patients (now 12 and 13 years old) received surgery when they were 1 and 2 years old.

         2c) in Fig. 4, the BRAF gene is erroneously reported twice as BARF.

         2d) Table 2 requires a reconfiguration since some sentences are invading the next line.

         2e) It is unclear to me whether the described pathogenic changes have been observed also in the lymphocytes from the patient blood or only in the brain tissue.

Author Response

Reviewer 2

The article genes-3549688 - "Identification of Novel Mosaic Variants in Focal Epilepsy Associated Patients’ Brain Lesions" by Bernardino García & coll.

reports about the genetic modifications underlining the human Focal Cortical Dysplasia Type III (FCDIII), as assessed by whole exome sequencing (WES). They report 10 novel pathogenic changes related to the expression of 8 genes. Some missense variants were considered to be highly pathogenic based on the estimated binding energy change.

The article is surely of high interest for future researches in all the fields of brain science. However, some attention should be given to the writing and to the data presentation:

1) In Materials and Methods:

1a) the authors states that 19 patients were studied, but in fig. 1 only 8 of them appear to have been studied by WES. How the remaining 11 patients contributed to the study?

Response: We appreciate the opportunity to clarify this discrepancy; Whole exome sequencing (WES) was performed on paired brain-blood samples from 19 FCDIII patients. After quality control and variant filtration (Figure 1A), All patients harboring high-confidence somatic variants were selected for detailed analysis (Table 1, Figure 2B). Of these, 8 patients exhibited pathogenic variants meeting functional and clinical thresholds (Figure 1B)

1b) there is NO definition of neither the P value that is the base of LogP, nor of log(q-value) both in Fig. 3B.  

Response: Thanks for your insight. In Figure 3B, -log10(P-value) represents the statistical significance of gene ontology terms (raw P-values from hypergeometric testing) These metrics were calculated using Metascape’s default pipeline, as described in their methodology Metascape, 2023. We modified this in the figure legend and Methods section.

1c) on line 140, how the "The detrimental effect on protein structure was measured in -kg/kcal according to the specified criteria" has been calculated and how it has been related to  "energy binding value" as reported in Fig.4 ?

Response: We apologize for the lack of clarity. In our study, DDMut computed the change in Gibbs free energy (ΔΔG, kcal/mol) to quantify the destabilizing effect of variants on protein structure. ΔΔG represents the difference in folding stability between wild-type and mutant proteins, with negative ΔΔG values (e.g., −2.5 kcal/mol) indicating destabilization (lower stability = higher pathogenicity). The term 'energy binding value' in Fig. 4 refers to these ΔΔG values. For example, a variant with ΔΔG = −1.25.0 kcal/mol (higher magnitude of destabilization) would correlate with a stronger pathogenic impact in ACY1. We have corrected '-kg/kcal' in the original text; this has been revised to 'kcal/mol' to align with DDMut’s standardized output.

2) In Results

2a) on line 8(156): the sentence "The mean age at the time of surgery was 4 years and 5 months." is confusing since on line 2(150) it is specified that "Epilepsy onset occurred at a median age of 1 year and 11 months..." The authors should specify that the sentence on line 8(156) is referring at the 8 patients that appear on Table 1.

Response: We sincerely apologize for the confusion. Upon re-examining the data for the 8 patients in Table 1, we would like to clarify the following: The mean age at surgery is 7 years and 10 months, calculated based on birthdates and surgery dates. Regarding epilepsy onset, the median age is 1 year and 3.5 months.

2b) on line 14(161), it is specified that "Cortical resection procedures were performed for the treatment of refractory epilepsy in 161 14 to 28-year-old participants at the time of referral (Table 1 &S4)" However, according to Table 1, the 28 year old patient received surgery in 2010, when he was 13, and the two youngest patients (now 12 and 13 years old) received surgery when they were 1 and 2 years old.

Response : We sincerely apologize for the inconsistency. Upon re-examining the data, the original statement on line 161 conflated age at referral with age at surgery. Using the provided dates of birth and surgery for the 8 patients, the corrected age range at the time of surgery is 1 to 13 years, as follows

2c) in Fig. 4, the BRAF gene is erroneously reported twice as BARF.

Response: We sincerely apologize for the typographical error in Figure 4. The misspelling of BRAF as BARF has been corrected in the revised manuscript. All instances now accurately reflect BRAF (B-Raf proto-oncogene, serine/threonine kinase), and the figure and legend have been updated accordingly. Thank you for identifying this oversight, which has been rectified to ensure clarity and scientific accuracy

2d) Table 2 requires a reconfiguration since some sentences are invading the next line.

Thank you for highlighting this formatting issue. We have reconfigured Table 2 to improve readability.

2e) It is unclear to me whether the described pathogenic changes have been observed also in the lymphocytes from the patient blood or only in the brain tissue.

We appreciate the opportunity to clarify. All the pathogenic somatic variants identified in our study were exclusively detected in brain tissue and absent in matched blood (germline) samples. This confirms their somatic (non-inherited) origin, as blood-derived DNA served as a control to filter out germline variants. For example, variants in BRAF or CNTNAP2 were found in brain biopsies but not in peripheral blood lymphocytes, supporting their role as tissue-specific drivers of FCDIII. We have revised the Methods section to explicitly state:
‘Somatic variants were defined as those present in brain tissue (allele frequency ≥5%) but undetectable in paired blood samples, ensuring the exclusion of germline or constitutional mutations